# Are Medical Graduates’ Job Choices for Rural Practice Consistent with their Initial Intentions? A Cross-Sectional Survey in Western China

**DOI:** 10.3390/ijerph16183381

**Published:** 2019-09-12

**Authors:** Jinlin Liu, Bin Zhu, Ning Zhang, Rongxin He, Ying Mao

**Affiliations:** 1Research Center for the Belt and Road Health Policy and Health Technology Assessment, Xi’an Jiaotong University, Xi’an 710049, China; binzhu2-c@my.cityu.edu.hk (B.Z.); ningzhang.xjtu@foxmail.com (N.Z.); herongxin@stu.xjtu.edu.cn (R.H.); 2Walter H. Shorenstein Asia-Pacific Research Center, Stanford University, Stanford, CA 94305, USA; 3School of Public Policy and Administration, Xi’an Jiaotong University, Xi’an 710049, China

**Keywords:** medical graduate, rural practice, China

## Abstract

Global concerns persist regarding the shortage and misdistribution of health workers in rural and remote areas. Medical education is an important input channel of human resources for health. This study aimed to identify the association between medical graduates’ job choices for rural practice and their initial intentions when they began to look for a job in China. Data were extracted from a cross-sectional survey among medical students in ten western provinces in China in 2013. Only medical students who were in the last year of study (i.e., medical graduates) and had found a job were included in this study. Of the 482 participants, 61.04% (293) presented an initial intention of rural practice when they began to look for a job, and 68.88% (332) made a final job choice for rural practice. However, of the 332 graduates with a final job choice of rural practice, only 213 (64.55%) had an initial intention. A univariate association was identified in which medical graduates who were more likely to make final job choices for rural practice were those having initial intentions (OR: 1.59; 95% CI: 1.08–2.36); however, after adjusting for controlled variables, it became insignificant and was reduced to a 1.31-fold increase (95% CI: 0.82–2.07). The initial intentions of medical graduates are not assurance of ultimate job outcomes, and it cannot be deduced that all medical graduates who made a final job choice for rural practice had authentic desires for rural practice. Twenty years of age or below, low-income families, majoring in non-clinical medicine, and studying in a junior medical college or below were associated with medical graduates’ final job choices for rural practice. More studies are required on how to translate medical student’s intention of rural medical practice into reality and how to retain these graduates via a job choice in rural practice in the future.

## 1. Introduction

The global Sustainable Development Goals (SDGs) were officially put forward in December 2015 by the United Nations, in which the health of populations have a central position because of its inalienable and non-negligible contributions to both societies and individuals [1]. Meanwhile, it is a universal truth that there is no health without a health workforce [2]. However, there are global concerns regarding the shortage and misdistribution of qualified health workforces, which affect almost all countries, especially relatively poor countries. Furthermore, rural and remote areas in countries are facing the most severe challenges in terms of the health workforce shortage and misdistribution. Similar challenges also exist in China. Approximately 43.90% of China’s population lives in rural areas [3]; however, only 37.70% of the health workforce is located there [4]. The uneven allocation of the health workforce between China’s urban and rural areas further weakens rural people’s access to the health workforce and results in poorer health among rural people [5,6]. Therefore, it has drawn a lot of attention from governments and researchers in terms of finding solutions which attract more health workers to work in rural areas.

As is known, medical students are the most important source of future health workers. However, although the number of medical graduates is increasing gradually, the shortage of health workers working in rural areas persists. A related systematic review reported that medical students studying in low-income or middle-income countries had low intentions (10–20%) regarding future careers working in rural locations [7], including medical students from Nepal [8], Bangladesh [9], sub-Saharan Africa [10], Ethiopia [11], South Africa [12], and India [13]. In addition, a cross-sectional survey conducted in five Asian countries found that approximately 60%, 60%, 50%, 50%, and 33% of medical students had a positive attitude towards future rural practice after graduation in Bangladesh, Thailand, China, India, and Vietnam, respectively [14]. Many factors effect medical students’ future career choices regarding rural or urban practice. Bowman et al. [15] posited that the low numbers of medical graduates presenting career intentions for rural practice was due to the fact of their lack of adequate understanding and contact with rural practitioners, people, and communities. Making the final decision for future career location is a difficult but very important process for medical graduates, which will affect their future working conditions, career development, family life, and social relationships [16]. Therefore, whether medical students’ intentions of rural practice ultimately translate into actual behavior of rural practice remains a matter of contention [17].

Many studies have been conducted to identify what factors affect the rural practice intentions of medical students. Kirschbaum et al. [18] reported that pharmacy medical students’ career intentions of rural practice may be positively affected by exposure to rural locations, such as a rural background, a rural placement, etc. By conducting a survey in five countries in Asia among medical students in their last year of study, Chuenkongkaew et al. [14] found that the influencing factors associated with medical students’ attitudes towards rural practice after graduation were gender and residence, including their parents’ residence and their residence during high school. A study [19] in China reported that both personal factors, such as gender and job concern, and some family factors were significantly related to medical students’ positive attitudes towards rural medical work. Another study [20] in Argentina found age, specialty, and prior experience serving in a deprived area to be significantly associated with medical students’ intentions of practicing medicine in rural or underprivileged areas. In addition, a large number of rural early exposure programs have been found to have a significant impact on increasing the interests or intentions of working in rural areas among medical students [21,22,23,24,25].

Meanwhile, some studies have reported the factors associated with medical graduates’ ultimately practicing in rural areas. Budhathoki et al. [7] established a conceptual framework to identify the motivations of medical graduates to work in rural areas after graduation, which consisted of personal and lifestyle, medical school, medical training and curriculum, health facility, and policy-related factors. A rural background (i.e., growing up in rural areas), studying at medical schools located in rural areas, and rural clinical clerkship or early exposure during medical training in rural areas under a community-based curriculum, etc. can motivate medical students to choose a rural medical institution for work after graduation [7,26]. Australian studies by Puddey et al. revealed that medical students’ gender, age, origin, type of registration, and type of medical school were significantly related to their choice to working in rural areas or areas with low socio-economic conditions after graduation [17,27]. Rural exposure was also reported to be an independent predictor of medical students’ future rural practice [26,27,28].

In addition, some studies analyzed the association between initial intention of rural practice and ultimately opting for rural practice among medical students; however, it is still a controversial issue. Herd et al. [29] reported that medical students who presented positive intentions of rural practice at entry were more likely to actually work in a rural area at postgraduate year (PGY) 1 (OR: 1.38; 95% CI: 1.01–1.88) or PGY 3 (OR: 1.86; 95% CI: 1.30–2.64) than those without an initial intention of rural practice. Another study [30] conducted in the United States reported that rural origin, rural intention, and general practice intention of medical students when they entered medical schools were significantly associated with rural practice 30 years after graduation. However, Playford et al. [31] pointed out that interest alone when medical students are studying in rural medical schools is not enough; they found that entry intention to practice rurally was not significantly related to the ultimate choice of rural practice when adjusting for other confounders, and the experience of rural clinical school participation of medical students was the deciding factor for realizing this intention [26].

Currently, no related studies have been reported and found to analyze the association between medical graduates’ job choices for rural practice and their initial intentions when they studied at schools or when they began to look for a job in China. Therefore, using the data extracted from a cross-sectional survey among medical students studying in western provinces of China, this study aimed to identify whether medical graduates’ job choices for rural practice were consistent with their initial intentions when they began to look for a job in China.

## 2. Materials and Methods

### 2.1. Study Design

This cross-sectional survey was one section included in the research project “*Situational Analysis and Policy Evaluation of Development and Retention of Human Resources for Health in Rural Western China*”. This collaborative research project was implemented by ten research teams in ten western provinces in China including Gansu, Kweichow, Inner Mongolia, Ningxia, Qinghai, Shaanxi, Sichuan, Tibet, Xinjiang, and Yunnan. Xi’an Jiaotong University in Shaanxi was the general coordinator of the project. The China Medical Board (CMB) provided funding for the project, and experts from the World Health Organization (WHO) provided technical support and project supervision [32].

Data used in this study were extracted from a cross-sectional survey implemented among senior medical students, i.e., medical students in third-, fourth-, or fifth-year grades, as they might have a preliminary or better understanding of their future career planning. Ten research teams conducted the survey in ten provinces at the same time. A two-stage random sampling method was utilized, and Figure 1 shows the sampling process. In terms of the sample size, all the ten co-PIs (Principle Investigators) in the ten research teams and two experts from the WHO made decisions together under a comprehensive consideration of sample representativeness and the amount of project funds. As Figure 1 shows, in each of the ten provinces in western China, six medical schools or fewer were first selected randomly, which included two technical secondary medical schools or fewer, two junior medical colleges or fewer, and two medical universities or fewer. Then, in each medical school, no more than 100 medical students were randomly selected and those who were willing to take part in survey were invited to fill in the questionnaire. Approximately 5000 medical students were selected in all ten provinces and 4517 medical students took part in the survey and filled in the questionnaire [32].

As this study focused on medical students who were in their last year of study (i.e., medical graduates) and had found a job, only the questionnaires that met such criteria were extracted. Generally, medical students in China start looking for and gradually find a job during their last year of study. According to this, 482 eligible participants in total were included in this study [33].

### 2.2. Data and Variables

As the general project coordinator, the research team from Xi’an Jiaotong University administrated all the procedures related to the questionnaire used for data collection. First, following the research objectives of the project, the team from Xi’an Jiaotong University designed an initial questionnaire. Then, each of other nine research teams from nine western provinces in China validated the questionnaire via group discussion based on their previous relevant research experience. Some research teams conducted small-scale pre-surveys with medical students to validate the logic and rationality of the questionnaire. Finally, all ten research teams revised and completed the questionnaire together. Meanwhile, the two WHO experts, (i.e., Fethiye Gulin Gedikg and Chunmei Wen), provided a significant amount of professional technical support during the process of questionnaire design. After the questionnaire was revised and finalized, the survey was conducted from June to December 2013 [32]. The participants filled in the questionnaire by themselves.

According to this study’s objectives, as the completed questionnaire contained many elements, only the relevant variables and data from medical graduates who had found jobs were extracted for analysis from the original dataset. These variables consisted of three sections, including basic sociodemographic indicators, medical graduates’ initial intentions for rural practice when they began to look for a job, and their current employment status.

Specifically, there were seven sociodemographic indicators [32] which included: (1) gender with two groups, i.e., male and female; (2) age, which was changed from a continuous variable to a binary variable with two groups, i.e., ≤20 years and ≥21 years; (3) residence, with two groups, i.e., rural and urban; (4) income level of the medical student’s family (monthly per capita income) with three groups, i.e., low level (<1000 Yuan), medium level (1000–4999 Yuan), and high level (≥5000 Yuan); (5) education level of the medical student’s parents with three groups, i.e., low level (primary school or below), medium level (junior high school), and high level (senior high school or above); (6) specialty with two groups, i.e., clinical medicine and non-clinical medicine (including general practice, public health, and other); (7) type of medical school with two groups, i.e., junior college or below (i.e., junior college or technical secondary school), and university (undergraduate).

The initial intention of rural practice when medical graduates began to look for a job was a binary variable (i.e., yes or no), which was identified via a dichotomous question, “*Are you willing to work in rural medical institutions after graduation when you begin to look for a job*?”.

The current employment option (job choice) for rural practice was also a binary variable (i.e., yes or no), which was identified according to the medical graduates’ responses to their job status.

### 2.3. Statistical Methods

The one-sample K–S test (i.e., one-sample Kolmogorov–Smirnov test) was used firstly to test the normality of the continuous variables. In our study, the variable of age was continuous originally, and it showed an abnormal distribution after test, so the “median” and “interquartile range (IQR)” were used to describe it. In addition, the categorical variables were displayed by “number” and “percentage”.

Pearson’s chi-squared tests were performed to identify the differences in the percentages of initial intention of rural practice and sociodemographic characteristics between medical graduates whose current employment options were rural practice and those who did not opt for rural practice. Related chi-squared values and the *p*-values are presented.

Binary logistic regression analyses were conducted to determine the association between medical graduates’ initial intentions for rural practice when they began to look for a job and their current job choices for rural practice. In the logistic regression models, medical graduate’s current job choice for rural practice was set as the dependent variable, medical graduate’s initial intention of rural practice was the independent variable, and the controlled variables included all seven of the sociodemographic indicators, i.e., gender, age, residence, income level of medical students’ family, education level of medical students’ parents, specialty, and type of medical school. The univariable logistic regression analyses were first performed to estimate the crude odds ratio (Cru. OR) with a 95% confidence interval (CI) for each of the independent variable and controlled variables. In addition, a multivariable logistic regression model was set up to further identify the influencing factors of the medical graduate’s current job choice for rural practice, and in this model, only the independent variable and controlled variables, which were significant in the univariable analyses, could be included. The adjusted odds ratio (Adj. OR) with a 95% CI is reported. The *p*-value < 0.05 was set as the significance level.

Meanwhile, we further separately identified the influencing factors related to medical graduates’ current job choices for rural practice for those who had an initial intention of rural practice when they began to look for a job and for those who did not have an initial intention.

For data analyses, the Statistical Package for Social Sciences 24.0 (SPSS, IBM, Armonk, New York, NY, USA) for MAC was used.

### 2.4. Ethics

The Ethics Committee of the School of Medicine of Xi’an Jiaotong University approved the study, and the approval number was 2014189. Our research strictly abided by the ethical principles of the Medical Ethics Committee although it did not involve human trials. The questionnaire used in the survey was anonymous and verbal informed consent was collected from all the participants. The data were protected according to the China Medical Board regulations on data protection and privacy guidelines, and the datasets used during the current study are available from corresponding author on reasonable request.

## 3. Results

### 3.1. Descriptive Analyses

As Figure 1 shows, 482 medical graduates who had found a job were included in the study. Table 1 reports the results related to participants’ sociodemographic characteristics, initial intentions of rural practice when they began to look for a job, and their current job choices for rural practice.

A total of 62.45% of the participants were female. The median age of these medical graduates was 22 years with the IQR as 21–23 years; meanwhile, more than four-fifths were older than 21 years. In terms of residence, approximately seven-tenths (68.88%) of participants came from rural areas. Most of the medical graduates came from a low-income family (38.75%) or a medium-income family (47.71%), i.e., the monthly per capita incomes of the medical graduates’ families were <1000 or 1000–4999 Yuan, respectively. In terms of the education of the participants’ parents, more than one-third of the medical graduates’ fathers (36.17%) and more than half of their mothers (54.77%) only attained a low-level education, i.e., primary school or below. Of the medical graduate participants, 88.57% majored in the specialty of clinical medicine and 11.43% were from other specialties, i.e., general practice, public health, etc. Furthermore, 53.53% of the participants were from junior medical colleges or technical secondary medical schools, followed by universities (46.47%). In terms of initial intention of rural practice, 61.04% (293/480) of the medical graduates disclosed initial intentions of rural practice when they began to look for a job.

In terms of the participants’ current job choices for rural practice, of the 482 medical graduates, 332 (68.88%) found a job in rural medical institutions.

### 3.2. Crosstab Analyses and Pearson’s Chi-Squared Tests

Table 2 displays the results of crosstab analyses and Pearson’s chi-squared tests. It shows that a significant difference was identified between medical graduates with current job choices for rural practice and those without a job choice for rural practice with respect to their initial intentions of rural practice when they began to look for a job (*p* < 0.05). Compared with medical graduates whose current job choices were not rural practice (53.33%), their counterparts with a job choice for rural practice presented a significantly higher proportion (64.55%) in the group of having an initial intention of rural practice when they began to look for a job.

Meanwhile, significant differences were observed between medical graduates whose current job choices were rural practice and those without a job choice for rural practice with respect to age (*p* < 0.01), residence (*p* < 0.001), income level of the medical graduates’ families (*p* < 0.001), education level of medical graduates’ parents (*p* < 0.001), specialty (*p* < 0.001), and type of medical school (*p* < 0.001). In comparison to medical graduates without current job choices for rural practice, these medical graduates whose current job choices were rural practice presented significantly higher proportions in the following characteristics: ≤20 years, rural residents, low- and medium-income families, low-level education of parents, majoring in a specialty of non-clinical medicine, and studying in junior medical colleges or below.

However, we did not observe a significant difference regarding the gender of medical graduates (*p* > 0.05).

### 3.3. Binary Logistic Regression Analyses

When conducting binary logistic regression analyses on medical graduates’ current job choices for rural practice, univariable logistic regressions were performed first, after which multivariable analyses were conducted. All results are displayed in Table 3.

Based on univariable logistic regressions, the estimation of crude OR with a 95% CI indicated that the medical graduates who were more likely to make a current job choice for rural practice were those who had an initial intention of rural practice when they began to look for a job (OR: 1.59, 95% CI: 1.08–2.36, compared with “having no initial intention”). Meanwhile, estimation of crude OR with 95% CI indicated that medical graduates who were more likely to make a current job choice for rural practice were those who were 20 years or below (OR: 2.44, 95% CI: 1.37–4.36, compared with “≥21 years”), those who were from rural areas (OR: 2.88, 95% CI: 1.91–4.33, compared with “urban”), those who were living in a low-income (OR: 6.88, 95% CI: 3.71–12.78, compared with “high-income”) or a medium-income family (OR: 4.42, 95% CI: 2.47–7.91, compared with “high-income”), those whose fathers or mothers only attained a low-level education (OR: 3.14, 95% CI: 1.86–5.28 for father; OR: 2.69, 95% CI: 1.63–4.43 for mother, compared with “high-level education”), those who majored in a specialty of non-clinical medicine (OR: 4.13, 95% CI: 1.73–9.86, compared with “clinical medicine”), and those who studied in the junior medical colleges or below (OR: 4.57, 95% CI: 3.01–6.96, compared with “university”).

After adjusting the independent variable and all the controlled variables which were significant in the univariable analyses in the multivariable logistic regression model (i.e., Model 1), the association of medical graduates’ initial intentions of rural practice when they began to look for a job with their current job choices for rural practice became insignificant (OR 1.31, 95% CI: 0.82–2.07, compared with “having no initial intention of rural practice”).

Age, income level of medical graduates’ families, specialty, and type of medical school were still significantly associated with medical graduates’ current job choices for rural practice. The medical graduates who were more likely to make current job choices for rural practice were those who were 20 years or below (OR: 2.51, 95% CI: 1.28–4.94, compared with “≥21 years”), those who came from low- or medium-income families (OR: 2.82, 95% CI: 1.30–6.14 for low-income; OR: 2.53, 95% CI: 1.31–4.88 for medium-income, compared with “high-income”), those who majored in a specialty of non-clinical medicine (OR: 4.86, 95% CI: 1.91–12.40, compared with “clinical medicine”), and those who studied in junior medical colleges or below (OR: 2.86, 95% CI: 1.67–4.91, compared with “university”). However, residence and education of the medical graduates’ parents were no longer significantly associated with medical graduates’ current job choices for rural practice.

Moreover, Table 4 presents the results of influencing factors associated with current job choices for rural practice among medical graduates with versus without initial intentions of rural practice when they began to look for a job separately. For medical graduates with initial intentions of rural practice, the age, residence, income level of medical graduates’ families, education level of medical graduates’ parents, specialty, and type of medical school were significantly associated with medical graduates’ current job choices for rural practice based on the univariable analyses; however, after introducing these significant variables in Model 2, only age, specialty, and type of medical school remained significant. Medical graduates who had an initial intention of rural practice and were more likely to make current job choices for rural practice were those who were 20 years or below (OR: 2.63, 95% CI: 1.03–6.74, compared with “≥21 years”), those who majored in a specialty of non-clinical medicine (OR: 3.92, 95% CI: 1.36–11.30, compared with “clinical medicine”), and those who were studying in junior medical colleges or below (OR: 2.16, 95% CI: 1.03–4.54, compared with “university”).

With respect to the medical graduates who did not have initial intentions of rural practice, residence, income level of the medical graduates’ families, education level of medical graduates’ parents, specialty, and type of medical school of the medical graduates were significantly associated with their current job choices for rural practice based on the univariable analyses. However, after conducting multivariate regression analysis, the results of Model 3 showed that only the income level of medical graduates’ families and type of medical school remained significant. Medical graduates who did not have an initial intention of rural practice and were more likely to make current job choices for rural practice were those who came from a low-income or a medium-income family (OR: 18.31, 95% CI: 3.34–100.38 for low-income; OR: 15.86, 95% CI: 3.21–78.44 for medium-income, compared with “high-income”), and those who studied in junior medical colleges or below (OR: 4.11, 95% CI: 1.79–9.43, compared with “university”).

## 4. Discussion

To the best of our knowledge, this study is the very first to analyze the association between medical graduates’ job choices for rural practice and their initial intentions when they began to look for a job in China. A total of 482 medical graduates extracted from a large-scale cross-sectional survey of medical students in western China participated in this study. It provides relevant experience and evidence from China for the growing body of studies and literature on medical students’ or graduates’ attitudes or perceptions towards working in rural areas and their actual career choices.

The results revealed that the association between medical graduates’ final job choices for rural practice and their initial intentions of rural practice when they began to look for a job was not as significantly strong as expected, which was in line with the results of Playford et al.’s [26,31] studies, but contradictory to Herd et al.’s [29] study. First, among all the 482 medical graduates, 293 (61.04%) had an initial intention of rural practice when they began to look for a job, and 332 (68.88%) ultimately made a job choice for rural practice. However, among the 332 medical graduates who ultimately opted for rural practice, only 64.55% (213) had initial intentions of rural practice. Second, after making logistic regression analyses, only a univariate association was determined between initial intentions of rural practice among medical graduates and their final job choices. A 1.59-fold increase was seen in the odds of making a final job choice for rural practice among medical graduates who had an initial intention; however, it was reduced to a 1.31-fold increase and was no longer statistically significant after adjusting for all of the controlled variables. These results indicated that the initial intention of rural practice among medical graduates cannot be an assurance of the ultimate outcome, and it also cannot be deduced that all medical graduates who had made a final job choice for rural practice had authentic desires for rural practice. In addition, it should be the focus of attention that the initial intention of rural practice among medical graduates in this study was not the entry intention, i.e., the intention when they first entered medical school which was measured in Playford et al.’s [26] study, but the intention when they began to look for a job and were ready to graduate from medical school. This might bias the results of this study compared with prior studies to some extent.

Our study found that residence was not a completely significant factor related to the medical graduates’ job choices for rural practice when controlling for other variables, which was inconsistent with previous studies [26,34]. However, residence was identified as a very important factor by several prior studies conducted in Botswana [35], Nepal [36], and England [11]. Although residence was not an independent influencing factor in our study, the results of the univariate logistic regression analysis indicated that having a rural background was significantly associated with medical graduates’ final job choices for rural practice; furthermore, medical graduates coming from rural areas were 2.88 times (crude OR) more likely to make a final job choice for rural practice compared with students coming from urban areas. Meanwhile, 76.20% of rural medical graduates in our study finally made a job choice for rural practice, which was significantly higher than the results (52.67%) of medical graduates with an urban background. In a systematic review study, the rural background of medical students was identified as the strongest factor which was associated with their rural practice [34]. In addition, Playford et al.’s [26] study also found that medical graduates who were coming from rural areas were nearly four times more likely to practice rurally than those urban origin graduates. Therefore, recruiting medical graduates coming from rural areas has been taken into account and implemented in many related policies and projects in countries aiming to improve the shortage of the health workforce in rural or remote areas [32]; for example, the ongoing programme named Rural-Oriented Tuition-Waived Medical Education (RTME) in China [37].

In addition, four influencing factors were found in our study to be associated with medical graduates’ final job choices for rural practice. The first factor was medical graduates’ age. Medical graduates with an age of 20 years or below were 2.51 times more likely to make a final job choice for rural practice compared with those graduates who were ≥21 years. Decreased age significantly increased the possibility for medical graduates to make a final job choice for rural practice. Compared with younger medical graduates, these older graduates are affected by more factors when they looked for a job; for example, they might have more needs and expectations to find a good job to support their families, and it is common that the average salary of healthcare workers and working conditions in urban medical institutions is higher and better than those in rural medical institutions. However, this result was contradictory to the prior studies of Puddey et al. [17] and Playford et al. [38] conducted in Australia, which reported that the older age of medical graduates was significantly related to their job choices of working in rural areas or areas with low socio-economic conditions. Similarly, Wayne et al. [39] also noted that older medical students were nearly twice more likely to work in the communities with underserved health care conditions.

The second factor was income, i.e., the monthly per capita income of a medical graduate’s family. Few studies have analyzed the association of family income with a medical graduate’s or medical student’s attitudes or career choices towards rural practice. In our study, medical graduates coming from lower-income families were 2.53–2.82 times more likely to ultimately choose a job in rural areas compared with those from high-income families. Lower family income levels of medical graduates significantly improved the possibility for them to make the final job choices for rural practice. Similar results were presented in another study by us, that showed when the income level of a medical graduate’s family was regular rather than unstable, they are more likely to exhibit negative attitudes towards rural medical work [37]. One possible explanation was that medical graduates from higher-income families might prioritize the monetary income from work and rural work cannot achieve their objectives [36].

The third factor was medical graduates’ specialties. A 4.86-fold increase was found in the odds of making final job choices for rural practice among medical graduates who majored in the specialty of non-clinical medicine such as general practice, public health, or others compared with those students who majored in the specialty of clinical medicine. This result was consistent with that in the study by Puddey et al. conducted in Australia, which reported that medical graduates who had a qualification in general practice were more likely to work in areas with low socioeconomic conditions [17]. In reality, primary medical institutions in China are short on public health personnel and general practitioners [32]; therefore, recruiting more medical graduates who major in non-clinical medicine, especially general medicine or public health, would be beneficial to some extent.

The type of medical school that medical graduates studied at was the fourth factor. Compared with medical graduates who studied in medical universities, meaning they could obtain a bachelor’s degree or above, those graduates who studied in junior medical colleges or below were 2.86 times more likely to make a final job choice for rural practice. In China, there are mainly two categories of medical degree education programs: the first is a 3-year medical training program with a diploma certificate, and the other is a medical university which provides bachelor’s and master’s degrees or doctorates under 5, 7, and 8 years or longer programs, respectively [40]. Another study by Qing et al. reported similar findings, where medical students who engage in a three-year program (junior college) are more likely to intend to work in rural areas [19]. It can be understood that after studying and graduating from a medical university where medical students have spent much more time, energy, and money for a high-level degree, they would give more priority to working in urban areas, even in a high-level urban hospital, where the working and living conditions are better than working in a rural area [32]. Meanwhile, medical graduates studying in junior colleges or below have little advantages when looking for a job in an urban area compared to those who studied in universities due to the fact of their lower level of medical education. In addition, health workers who graduate from a junior medical college or below and receive a low-level medical degree are basically competent to work in primary medical institutions in rural areas.

Our study demonstrated the differences in influencing factors associated with current job choices for rural practice among medical graduates with versus without initial intentions of rural practice when they began to look for a job. The type of medical school was the only significant factor associated with medical graduates’ ultimately opting for rural practice, no matter whether they presented initial intentions of working in rural areas or not. Medical graduates who studied in a junior medical college or below were 2.16 times and 4.11 times more likely to ultimately find a job in a rural area for those with an initial intention and those without an initial intention, respectively. In addition, age and specialty were two significant factors influencing the final choice for rural practice among medical graduates who already had initial intentions of future rural work, and the income level of medical graduates’ families was the significant factor associated with ultimately opting for rural practice among medical graduates without an initial intention.

Meanwhile, three other sociodemographic variables, including gender, education level of medical graduates’ fathers, and education level of medical graduate’s mothers, were not significantly associated with medical graduates’ initial intentions of rural practice or ultimately opting for rural practice. Gender is still a controversial factor. Similar results were reported by Puddey et al. [17] and Borracci et al. [20] regarding the insignificant effect of gender; however, Huntington et al. [8], Qing et al. [19], and Van Wyk et al. [12] demonstrated that, compared with female medical students, male medical students were more likely to ultimately practice in rural areas in Nepal, China, and South Africa, respectively. In our study, 72.4% of male graduates ultimately found a job in rural areas, which was higher than that (66.8%) of female medical graduates. In terms of the education level of medical graduates’ parents, only a univariate association was observed between the education level of the parents and medical graduates’ ultimately opting for rural practice, and a low-level education of medical graduates’ parents increased the odds of medical graduates’ ultimately opting for a job in rural areas.

This study has a few implications. Although the independent effect was not identified, our study still highlighted the importance of medical graduates’ or medical students’ initial intentions of rural practice to some extent. Only these who have the will to work in rural areas are more likely to choose a job in rural medical institutions; after they experience rural work, they will also be more likely to be involved in the delivery of rural healthcare services. Thus, to identify key influencing factors and implement interventions to improve medical students’ intentions for rural practice during their study in medical schools, more attention should be paid by related policy makers, development agencies, and medical schools in China. Meanwhile, the importance of the rural background of medical students cannot be ignored, and recruiting more medical students from rural areas should still be an effective strategy for improving the rural health worker shortage. In addition, considering that majoring in non-clinical medicine and studying in a junior medical college or below are two significant factors that increase medical students’ odds of ultimately choosing a job in rural areas, these medical students might wish to be recruited by rural medical institutions in higher numbers.

Some limitations existing in this study should be acknowledged. First, in this cross-sectional study, the participants had found a job when the survey was conducted and they were asked to recall their initial intentions which would inevitably bring recall biases, i.e., the initial intention of rural practice among medical graduates could not exactly reflect their true attitudes at the time when they began to look for a job. Thus, all the analytical results related to intentions cannot be considered as conclusive. A cohort study would be much better and more accurate. Second, some longitudinal studies reported that medical graduates are highly mobile when they recently enter the rural workforce [41], and our study was a cross-sectional type which focused on a single point in time, so the student’s current option for rural practice was not necessarily long-term choice. Third, because the sample size was not very large and participants were from western medical schools, conclusions of this study might not apply well to medical students in other regions. Fourth, because of the limited variables or indicators set in the questionnaire, many other factors which have been identified by previous studies were not included and analyzed. Fifth, all associations identified by our study cannot be concluded as causal relationships, because our study was of a cross-sectional type [32].

## 5. Conclusions

This study demonstrated a univariate association between medical graduates’ initial intentions of rural practice when they began to look for a job and their final job choices for rural practice in China. Based on a univariate analysis, a 1.59-fold increase was observed in the odds of finally making job choices for rural practice among medical graduates with initial intentions of rural practice when they began to look for a job. When adjusting for other possible predictors, although a 1.31-fold increase was seen, the association became statistically insignificant. The initial intention of rural practice among medical graduate cannot be an assurance of the ultimate outcomes, and it also cannot be deduced that all medical graduates who made a final job choice for rural practice had authentic desires for rural practice. In addition, age, income-level of medical graduates’ families, specialty, and type of medical school were significant influencing factors; specifically, 20 years of age or below, low-income families, majoring in non-clinical medicine, and studying in a junior medical college or below significantly increased the possibility of making a final job choice for rural practice. Further studies are required in order to understand how to translate medical students’ intentions of rural medical work into reality and how to retain these medical graduates via a job choice in rural practice in the future.

## Figures and Tables

**Figure 1 ijerph-16-03381-f001:**
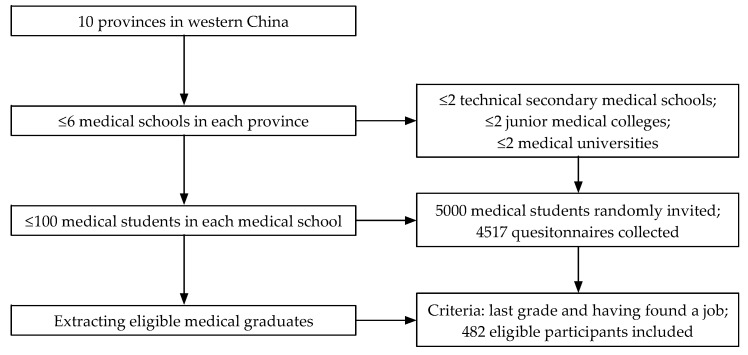
Study profile.

**Table 1 ijerph-16-03381-t001:** Participants’ sociodemographic characteristics, initial intentions of rural practice, and current job choices for rural practice.

Characteristics	Total *N* (%)	Characteristics	Total *N* (%)
Gender (*n* = 482)		Age (years, *n* = 482)	
Female	301 (62.45)	≤20	91 (18.88)
Male	181 (37.55)	≥21	391 (81.12)
Residence (*n* = 482)		Income (monthly per capita income of medical graduates’ families, Yuan, *n* = 480)	
Urban	150 (31.12)	<1000	186 (38.75)
Rural	332 (68.88)	1000–4999	229 (47.71)
		≥5000	65 (13.54)
Education of medical graduates’ fathers (*n* = 481)		Education of medical graduates’ mothers (*n* = 482)	
Low	174 (36.17)	Low	264 (54.77)
Medium	179 (37.21)	Medium	124 (25.73)
High	128 (26.61)	High	94 (19.50)
Specialty (*n* = 481)		Type of medical school (*n* = 482)	
Clinical medicine	426 (88.57)	Junior college or below	258 (53.53)
Non-clinical medicine	55 (11.43)	University	224 (46.47)
Initial intention of rural practice when participants began to look for a job (*n* = 480)		Current job choice for rural practice (*n* = 482)	
No	187 (39.96)	No	150 (31.12)
Yes	293 (61.04)	Yes	332 (68.88)

**Table 2 ijerph-16-03381-t002:** Crosstab analyses and Pearson’s chi-squared tests.

Characteristics	Current Job Choice for Rural Practice		Chi-Squared Test	
	Yes, *n* (%)	No, *n* (%)	Value	*p*-Value
Initial intention of rural practice when participants began to look for a job (*n* = 480)			5.45	0.021
No	117 (35.45)	70 (46.67)		
Yes	213 (64.55)	80 (53.33)		
Gender (*n* = 482)			1.65	0.223
Female	201 (60.54)	100 (66.67)		
Male	131 (39.46)	50 (33.33)		
Age (years, *n* = 482)			9.59	0.002
≤20	75 (22.59)	16 (10.67)		
≥21	257 (77.41)	134 (89.33)		
Residence (*n* = 482)			26.71	<0.001
Urban	79 (23.80)	71 (47.33)		
Rural	253 (76.20)	79 (52.67)		
Income (monthly per capita income of medical graduates’ families, Yuan, *n* = 480)			43.54	<0.001
<1000	147 (44.28)	39 (26.35)		
1000–4999	162 (48.80)	67 (45.27)		
≥5000	23 (6.93)	42 (28.38)		
Education of medical graduates’ fathers (*n* = 481)			21.31	<0.001
Low	142 (42.77)	32 (21.48)		
Medium	115 (34.64)	64 (42.95)		
High	75 (22.59)	53 (35.57)		
Education of medical graduates’ mothers (*n* = 482)			21.22	<0.001
Low	205 (61.75)	59 (39.33)		
Medium	74 (22.29)	50 (33.33)		
High	53 (15.96)	41 (27.33)		
Specialty (*n* = 481)			11.70	<0.001
Clinical medicine	283 (85.24)	143 (95.97)		
Non-clinical medicine	49 (14.76)	6 (4.03)		
Type of medical school (*n* = 482)			54.11	<0.001
Junior college or below	215 (64.76)	43 (28.67)		
University	117 (35.24)	107 (71.33)		

**Table 3 ijerph-16-03381-t003:** Binary logistic regression on medical graduates’ current job choices for rural practice.

Variable	Univariable Logistic Regressions		Multivariable Logistic Regression (Model 1)	
	Cru. OR ^a^ (95% CI)	*p*-Value	Adj. OR ^b^ (95% CI)	*p*-Value
Initial intention of rural practice				
No	1		1	
Yes	1.59 (1.08–2.36)	0.020	1.31 (0.82–2.07)	0.257
Gender				
Female	1			
Male	1.30 (0.87–1.95)	0.199		
Age (years)				
≤20	2.44 (1.37–4.36)	0.002	2.51 (1.28–4.94)	0.008
≥21	1		1	
Residence				
Urban	1		1	
Rural	2.88 (1.91–4.33)	<0.001	1.39 (0.81, 2.37)	0.229
Income (Yuan)				
<1000	6.88 (3.71–12.78)	<0.001	2.82 (1.30–6.14)	0.009
1000–4999	4.42 (2.47–7.91)	<0.001	2.53 (1.31–4.88)	0.006
≥5000	1		1	
Education of medical graduates’ fathers				
Low	3.14 (1.86–5.28)	<0.001	1.54 (0.80–3.00)	0.199
Medium	1.27 (0.80–2.02)	0.315	0.97 (0.56–1.69)	0.925
High	1		1	
Education of medical graduates’ mothers				
Low	2.69 (1.63–4.43)	<0.001	1.12 (0.57–2.20)	0.740
Medium	1.15 (0.67–1.97)	0.625	0.85 (0.45–1.61)	0.618
High	1		1	
Specialty				
Clinical medicine	1		1	
Non-clinical medicine	4.13 (1.73–9.86)	0.001	4.86 (1.91–12.40)	0.001
Type of medical school				
Junior college or below	4.57 (3.01–6.96)	<0.001	2.86 (1.67–4.91)	<0.001
University	1		1	

^a^ Cru. OR: crude odds ratio. ^b^ Adj. OR: adjusted odds ratio.

**Table 4 ijerph-16-03381-t004:** Binary logistic regression on current job choices for rural practice among medical graduates (MGs) with versus without an initial intention.

Variable	MGs with an Initial Intention		MGs without an Initial Intention	
	Uni. ^a^	Multi. ^b^ (Model 2)	Uni. ^a^	Multi. ^b^ (Model 3)
	Cru. OR ^c^ (95% CI), *p*-Value	Adj. OR ^d^ (95% CI), *p*-Value	Cru. OR ^c^ (95% CI), *p*-Value	Adj. OR ^d^ (95% CI), *p*-Value
Gender				
Female	1		1	
Male	1.25 (0.73–2.15), 0.423		1.38 (0.75–2.56), 0.304	
Age (years)				
≤20	2.87 (1.24–6.66), 0.014	2.63 (1.03–6.74), 0.044	2.13 (0.94–4.84), 0.070	
≥21	1	1	1	
Residence				
Urban	1	1	1	1
Rural	2.66 (1.55–4.57), <0.001	1.54 (0.74–3.20), 0.251	3.31 (1.74–6.27), <0.001	1.40 (0.60–3.28), 0.443
Income (Yuan)				
<1000	4.56 (2.11–9.85), <0.001	1.68 (0.62–4.57), 0.312	31.58 (6.75–147.66), <0.001	18.31 (3.34–100.38), 0.001
1000–4999	2.90 (1.43–5.85), 0.003	1.63 (0.73–3.64), 0.236	18.97 (4.14–86.85), <0.001	15.86 (3.21–78.44), 0.001
≥5000	1	1	1	1
Education of medical graduates’ fathers				
Low	2.11 (1.05–4.22), 0.035	0.95 (0.38–2.36), 0.906	5.00 (2.22–11.28), <0.001	2.81 (0.99–7.89), 0.050
Medium	1.00 (0.53–1.88), 0.993	0.70 (0.33–1.47), 0.344	1.48 (0.72–3.05), 0.285	1.43 (0.59–3.46), 0.430
High	1	1	1	1
Education of medical graduates’ mothers				
Low	2.82 (1.46–5.43), 0.002	1.71 (0.73–4.03), 0.217	2.57 (1.17–5.64), 0.018	0.50 (0.16–1.62), 0.250
Medium	1.14 (0.56–2.32), 0.724	0.96 (0.43–2.14), 0.925	1.13 (0.48–2.64), 0.787	0.76 (0.25–2.33), 0.634
High	1	1	1	1
Specialty				
Clinical medicine	1	1	1	
Non-clinical medicine	3.36 (1.28–8.87), 0.014	3.92 (1.36–11.30), 0.012	4.99 (0.61–40.79), 0.134	
Type of medical school				
Junior college or below	3.41 (1.99–5.82), <0.001	2.16 (1.03–4.54), 0.042	7.02 (3.46–14.24), <0.001	4.11 (1.79–9.43), 0.001
University	1	1	1	1

^a^ Uni.: univariable logistic regression analyses. ^b^ Multi.: multivariable logistic regression analyses. ^c^ Cru. OR: crude odds ratio. ^d^ Adj. OR: adjusted odds ratio.

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
