# Peer review of "Are Medical Graduates’ Job Choices for Rural Practice Consistent with their Initial Intentions? A Cross-Sectional Survey in Western China"

_ijerph, 2019, doi:10.3390/ijerph16183381_

Round 1

Reviewer 1 Report

Congratulations to the originality and quality of the article.

In the item data and variables:
what was the reason they used the cut off point for age at 21 years.
In the definition of echelons (
low level (< 1000 Yuan), medium level (1000-4999 Yuan), and high level (≥ 5000 Yuan)), for the income levels used, have any national references?

In the tables 2 and 3:
In subtitles, suggested changing: COR for cOR and AOR for aOR ou Adj. OR, for example.

In Pag. 9, line 14:
Suggested to change: "became insignificant" for "not statistically significant".

In Pag. 9, line 30:
The value presented is not that of the adjusted model, and for which the difference is not statistically significant.

Author Response

Dear reviewer,

Many thanks for your review. Your comments and suggestions are very valuable to us, which contribute substantially to the refinement and improvement of our manuscript. We have studied them very carefully and try our best to revise the manuscript. A very detailed response to each comment and suggestion from you has been made. Please see the attached response letter for more details.

Reviewer 2 Report

Dear authors,

This is a very good article concerning reasons to redistribute population needs and medical services for rural areas.

As the article states, this is a very common problem in majority of countries worldwide and the issue must be discussed further to make possible new policies strategies relating scientific data and social development.

- Manuscript pages are repeating line numbers. Well anyway it is possible to reference a sentence by using page number and line number.

- References are not in the template form. It should be used brackets like [5].

- Abstract: I couldn't understand this line  "Of the 482 participants, 61.0% (293) and 68.9% (332) presented initial intentions when 19 they began to look for a job and made final job choices for rural practice, respectively; in the latter 20 group, 213 of 332 disclosed initial intentions."

- There is something strange in these data: Check the sentence " In terms of initial intention of rural practice, 61.0% (293/480) of the medical 32 graduates disclosed the initial intentions of rural practice when they began to look for a job." Then check table 1 last information in the table: 

Initial intention of rural practice when participants began to look for a job (n = 480)
No 187 (39.0) 117 (35.5) 70 (46.7)
Yes 293 (61.0) 213 (64.5) 80 (53.3)

If you compare the first information of table 1, gender, we can understand how many male and female have in the sample, and later how many said yes and no. But in the "initial intention of rural practice" we have already a definition of NO and YES with sample and later again a YES and NO filter.

I couldnt understand it well. For example: from 187 people who doesnt have initial intention, 117 have initial intention and 70 doesnt. Is this right?

I cant see the difference between filter of first YES-NO and second YES-NO.

Could you please explain it better dear authors?

I need a second review in order to track this information suggested for correction or better explanation. Some points in discussion, abstract and data are very interconnected with the starting affirmation "Of the 482 participants, 61.0% (293) and 68.9% (332) presented initial intentions when 19 they began to look for a job and made final job choices for rural practice, respectively; in the latter 20 group, 213 of 332 disclosed initial intentions."

Author Response

(The authors gave the same response as above.)

Reviewer 3 Report

Thank you for letting me review this paper.

It is advised to review the paper and delete the personal form “we…..”

The procedure for administering the questionnaires has not been described. Who administered them? How it was made?

What is 1-Sample K-S test?

There are unclear questions in ethical aspects such as: confidentiality, ethical principles, data protection...etc.

Results section: This section is ambiguous and it generates confusion. The results of the tables are mixed in the different headings. The author should give another meaning to this section or simply change the name of the headings.

Tables:

-Non-zero p-values should be stated, since p-values cannot be zero. More specifically, it is better to state p˂0.001 instead of p=0.000.

-One, two or three decimals appear. Unify data with two decimals.

-Table 3, Where are p-values?

Author Response

(The authors gave the same response as above.)

Round 2

Reviewer 2 Report

Very good modifications were done!

The article have improved a lot compared with the previous version.

Well done!

Reviewer 3 Report

Suggestions have been made.

This manuscript is a resubmission of an earlier submission. The following is a list of the peer review reports and author responses from that submission.